# Increase of cell surface vimentin is associated with vimentin network disruption and subsequent stress-induced premature senescence in human chondrocytes

**Jana Riegger\*, Rolf E Brenner**

Division for Biochemistry of Joint and Connective Tissue Diseases, Department of Orthopedics, University of Ulm, Ulm, Germany

**Abstract** Accumulation of dysfunctional chondrocytes has detrimental consequences on the cartilage homeostasis and is thus thought to play a crucial role during the pathogenesis of osteo-arthritis (OA). However, the underlying mechanisms of phenotypical alteration in chondrocytes are incompletely understood. Here, we provide evidence that disruption of the intracellular vimentin network and consequent phenotypical alteration in human chondrocytes results in an external-ization of the intermediate filament. The presence of the so-called cell surface vimentin (CSV) on chondrocytes was associated with the severity of tissue degeneration inclinical OA samples and was enhanced after mechanical injury of cartilage tissue. By means of a doxorubicine-based in vitro model of stress-induced premature senescence (SIPS), we could confirm the connection between cellular senescence and amount of CSV. Although siRNA-mediated silencing of CDKN2A clearly reduced the senescent phenotype as well as CSV levels of human chondrocytes, cellular senes-cence could not be completely reversed. Interestingly, knockdown of vimentin resulted in a SIPS-like phenotype and consequently increased CSV. Therefore, we concluded that the integrity of the intracellular vimentin network is crucial to maintain cellular function in chondrocytes. This assump-tion could be confirmed by chemically-induced collapse of the vimentin network, which resulted in cellular stress and enhanced CSV expression. Regarding its biological function, CSV was found to be associated with enhanced chondrocyte adhesion and plasticity. While osteogenic capacities seemed to be enhanced in chondrocytes expressing high levels of CSV, the chondrogenic potential was clearly compromised. Overall, our study reinforces the importance of the vimentin network in main-tenance of the chondrogenic phenotype and introduces CSV as a novel membrane-bound marker of dysfunctional chondrocytes.

**\*For correspondence:**
jana.riegger@uni-ulm.de

**Competing interest:** The authors declare that no competing interests exist.

## Editor's evaluation

The manuscript provides evidence for an association between cell surface vimentin (CSV) and chon-drocyte senescence. The authors have provided solid in vitro evidence about the role of cell surface vimentin (CSV) in chondrocyte senescence and its potential in OA development, although it is lacking in vivo evidence to prove the conclusion. Overall, this is well suited for a broad audience and is important for the OA/chondrocyte biology community.

## Introduction

Osteoarthritis (OA) is the most common joint disease worldwide and associated with various pathophysiological processes, including oxidative stress, cell death, inflammation, and catabolism. The presence of inflammatory cytokines, reactive oxygen species (ROS), and damage-associated molecular patterns create a harmful pathophysiological environment, which leads to cellular damage and phenotypical alteration of chondrocytes (*Riegger and Brenner, 2020*). By means of single-cell RNA sequencing, researchers could provide evidence for the accumulation of dysfunctional chondrocyte subtypes in OA cartilage, exhibiting a senescent, hypertrophic, or fibroblast-like phenotype (*Ji et al., 2019*; *Chou et al., 2020*). Phenotypical alteration of chondrocytes and subsequent dysfunctional behavior plays a crucial role in progressive cartilage degeneration. In fact, the secretome of OA chondrocytes contains excessive amounts of cytokines and chemokines, including IL-6, IL-8, and CXCL1, as well as catabolic enzymes, such as MMPs and ADAMTS (*Riegger and Brenner, 2020*). In case of senescent chondrocytes, the secretion of detrimental proteins is also referred to as senescence-associated secretory phenotype (SASP) (*Jeon et al., 2018*). SASP factors not only drive ongoing destruction of the extracellular matrix (ECM), but might also affect neighboring cells, in a paracrine manner. This leads to the activation of immunocompetent cells and subsequent inflammation, on one hand, and spreading of the senescent phenotype among chondrocytes, on the other hand (*Jeon et al., 2018*; *Vinatier et al., 2018*).

Dysfunctional and senescent chondrocytes have emerged as a central target in OA therapy. Accordingly, clearance of senescent chondrocytes has been found to attenuate the risk of injury-induced OA (*Jeon et al., 2017*). However, identification, characterization, and therapeutic targeting of senescent chondrocytes remains challenging due to the high heterogeneity in cellular senescence (*Kirschner et al., 2020*). Therefore, further cell type-specific investigation of the senescence-associated (SA) characteristics as well as the underlying mechanisms involved in senescence are needed.

Recently, an oxidized form of membrane-bound vimentin, namely malondialdehyde (MDA)-vimentin, was identified on senescent fibroblasts (*Frescas et al., 2017*). Initially, CSV has been considered as a marker of cancer stem-like cells, exhibiting epithelial-mesenchymal transition (EMT) characteristics (*Mitra et al., 2015*). However, current studies imply that CSV might also be expressed on non-cancer cells, such as bone marrow-derived mesenchymal stem cells (bmMSC) (*Ise et al., 2019*) and apoptotic neutrophils (*Moisan and Girard, 2006*). Despite increasing interest in the dislocated intermediate filament, the role and biological function of CSV is only poorly understood. Interestingly, disassembly of the intracellular vimentin network impaired chondrogenic characteristics in chondrocytes and thus contributes to an OA phenotype, as proposed by Blain et al. (*Tolstonog et al., 2001*; *Blain et al., 2006*). Since there are no reports about a potential link between the intracellular vimentin network and CSV, we wondered whether CSV can be found on human articular chondrocytes (hAC) and if the dislocation results from vimentin network disruption and consequent phenotypical alteration of cartilage cells.

Here, we provide first evidence of enhanced CSV on senescent hAC. CSV was associated with the severity of OA and was triggered by mechanical injury of cartilage tissue or doxorubicine (Doxo) stimulation and subsequent stress-induced premature senescence (SIPS). Interestingly, CSV was enhanced in response to the disruption of the intracellular vimentin network and ROS accumulation in vitro, indicating a link between intracellular integrity of vimentin, cellular stress, and the externalization of the intermediate filament. Furthermore, the results imply that CSV is associated with chondrocyte plasticity and dedifferentiation, underlining the importance of the vimentin network in the stabilization of the chondrogenic phenotype. Overall, this is the first report of CSV as novel membrane-associated indicator of chondrocyte plasticity and senescence.

## Results

### CSV and SA markers are increased in chondrocytes from highly degenerated or traumatized cartilage

In order to investigate the incidence of chondrosenescence and CSV-positive cells in OA, we compared hAC isolated from macroscopically intact (OARSI grade ≤1) and highly degenerated human cartilage tissue (OARSI grade ≥3) derived from the same patient. Exemplary characterization of the corresponding tissues using Saf-O staining confirmed clear differences regarding the proteoglycan content,

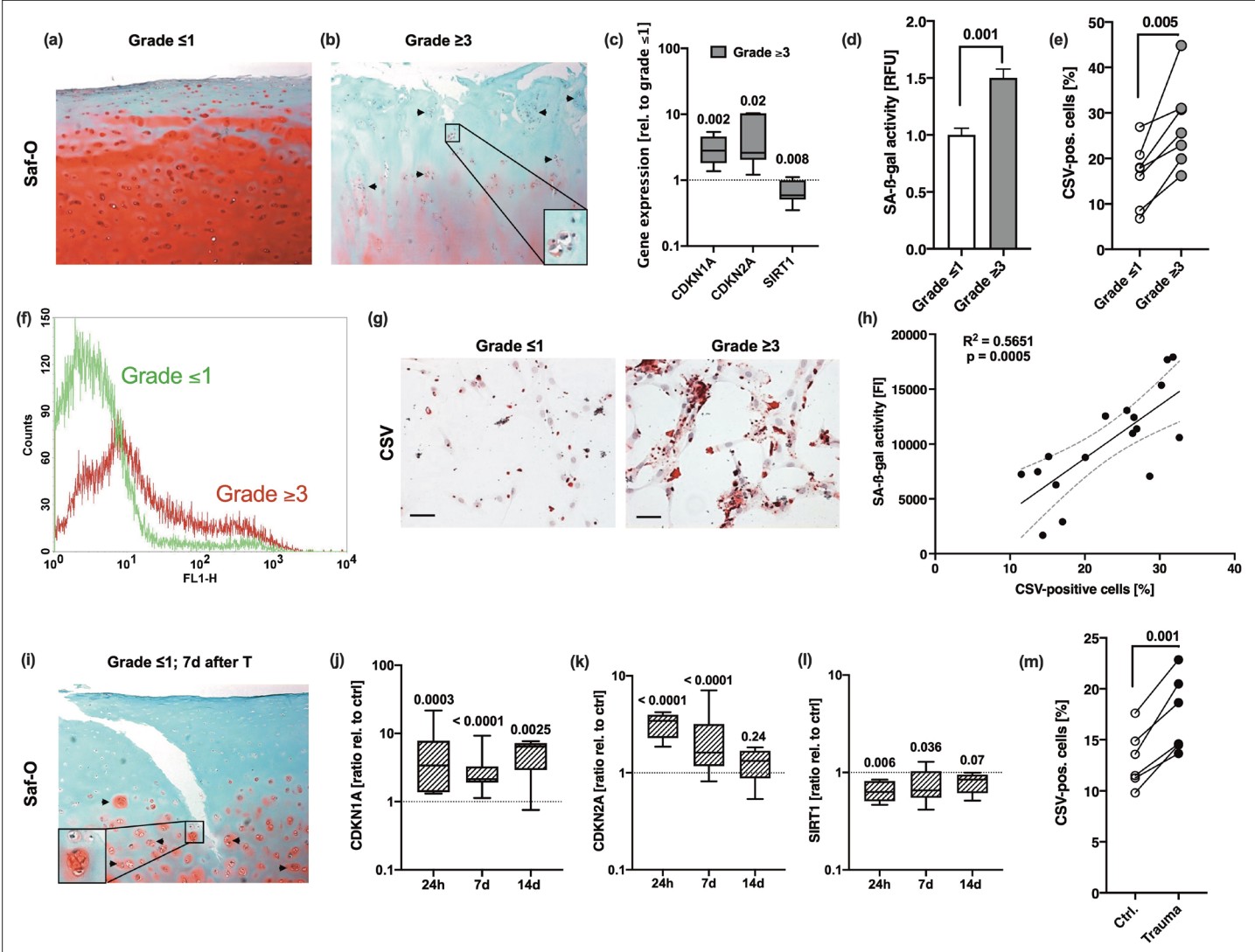

**Figure 1.** Association between chondrosenescence and the OARSI grade as well as mechanical injury. Safranin-O staining of (**a**) macroscopic intact (OARSI grade ≤1) and (**b**) highly degenerated (OARSI grade ≥3) cartilage tissue; scale bar: 200 μm. Cell clusters are indicated by black arrows. (**c**) Gene expression of CDKN1A, CDKN2A, and SIRT1 in highly degenerated cartilage (grade ≥3) relative to macroscopic intact (grade ≤1) tissue (n≥5). (**d–g**) Human articular chondrocytes (hAC) were isolated from macroscopic intact (grade ≤1) and highly degenerated (grade ≥3) cartilage tissue of same donors (matched pairs). (**d**) Senescence-associated-β-galactosidase (SA-β-gal) activity assay and (**e**) flow cytometric analysis of cell surface vimentin (CSV) (**f**) including an exemplary histogram demonstrating the shift between cells derived from grade ≤1 (green curve) and grade ≥3 (red curve) tissue. (**g**) Exemplary images of anti-CSV staining (immunocytochemistry) on isolated hAC derived from grade ≤1 and grade ≥3 tissue; scale bar: 50 μm. (**h**) Corresponding Pearson correlation analysis between percentage of CSV-positive cells and SA-β-gal activity. (**i**) Safranin-O staining of macroscopic intact (OARSI grade ≤1) cartilage 7 days after ex vivo traumatization. (**j–l**) Gene expression of (**j**) CDKN1A, (**k**) CDKN2A, and (**l**) SIRT1 in cartilage tissue (OARSI grade ≤1) 24 hr, 7 days, and 14 days after impact (n≥6); normalized to unimpacted cartilage. (**m**) CSV-positivity of hAC isolated from impacted (Trauma) or unimpacted (Ctrl) cartilage explants from the same donor at day 7 and analyzed after short in vitro cultivation (7 days, passage 0). Data are presented as box plots with median, whiskers min to max or scatter plots; or column bars with mean, scanning electron microscopy (SEM). Statistics: (**c**) was analyzed by a multiple t test; (**d**), (**e**), and (**m**) were analyzed by means of paired two-tailed t test; in case of (**j–l**) Kruskal-Wallis with a Dunn's post hoc test was used and (**h**) was analyzed by means of a Pearson correlation analysis. Each data point represents an independent biological replicate (**n**).

structural surface integrity, and cellularity between macroscopically intact and highly degenerated tissue (*Figure 1a and b*). mRNA levels of CDKN1A and CDKN2A were significantly enhanced in OARSI grade ≥3 as compared to OARSI grade ≤1 tissue, while that of SIRT1 was decreased (*Figure 1c*). SIRT1 has been described as an essential negative regulator in cellular senescence and was found to protect chondrocytes against IL-17-induced senescence (*Lee et al., 2019*; *Wang et al., 2021*).

Moreover, SA-β-galactosidase (SA-β-gal) and CSV were significantly higher in isolated hAC derived from highly degenerated cartilage as compared to hAC from donor-matched macroscopically intact tissue (*Figure 1d–g*). Taken together, increased activity of SA-β-gal positively correlated with the percentage of CSV-positive hAC (*Figure 1h*).

Our ex vivo human cartilage trauma model is suitable to study injury-induced oxidative stress, cell death, and degenerative processes (*Riegger and Brenner, 2020*; *Riegger and Brenner, 2019*; *Riegger et al., 2016*). Since ROS and cellular damage, in general, have been described as crucial triggers in chondrosenescence (*Jeon et al., 2018*; *Jeon et al., 2017*), we investigated the expression of SA markers and CSV on hAC after ex vivo traumatization. Seven days after traumatization, we could histologically confirm structural and cellular alteration of the formerly macroscopically intact cartilage, similarly to that of highly degenerated samples (*Figure 1i*). The traumatized tissue exhibited fissures in the surface, proteoglycan loss, hypocellularity, and cluster formation – altogether typical hallmarks of OA (*Riegger et al., 2018b*; *Riegger et al., 2018a*; *Lotz et al., 2010*). Moreover, the expression pattern of *CDKN1A*, *CDKN2A*, and *SIRT1* in traumatized cartilage explants was comparable to that in OARSI grade ≥3 tissue (*Figure 1j–l*). While mRNA levels of CDKN1A remained elevated over 14 days after cartilage trauma, that of CDKN2A and SIRT1 were largely restored to the control levels by time. Moreover, CSV was significantly higher on hAC derived from traumatized cartilage as compared to donor-matched hAC from unimpacted tissue (*Figure 1m*).

## Doxo-mediated SIPS enhances CSV levels on hAC

To verify that CSV was associated with a senescent phenotype of hAC, we used our Doxo-based in vitro model to induce SIPS in isolated hAC (*Kang et al., 2019*; *Kirsch et al., 2022*). As expected, Doxo treatment resulted in enhanced SA marker expression as well as SA-β-gal activity (*Figure 2a and b*), which was accompanied by a significant increase in CSV (*Figure 2c and d*).

For further investigation of the flow cytometric data, we defined four regions. Q1: CSV$^+$/SSC-H <400 (normal granularity). Q2: CSV$^+$/SSC-H >400 (unusually high granularity). Q3: CSV$^-$/SSC-H <400. Q4: CSV$^-$/SSC-H >400. This way, we found significant changes in the distribution to the four regions in hAC after Doxo stimulation as compared to the untreated control, characterized by a shift of the main population from SSC-H <400 to SSC-H >400, indicating an enhanced granularity, but also significantly increased numbers of CSV-positive cells, located in Q1 and Q2 (*Figure 2e–g*). Moreover, Doxo treatment resulted in morphological alteration of hAC, which was comparable to hAC isolated from highly degenerated cartilage (*Supplementary file 1*). Development of an abnormal fibroblast-like or flattened cell shape as well as increased granularity in Doxo-treated hAC has been previously described (*Kirsch et al., 2022*).

To visualize enhanced CSV levels on Doxo-treated hAC and further investigate whether there are similarities in the binding sites between the intracellular VIM antibody [clone EPR3776] and the CSV antibody [clone 84–1], non-permabilized hAC were co-stained with the corresponding antibodies (*Figure 2h*). In line with the cytometric analysis, immunofluorescence staining confirmed increased CSV levels on hAC after Doxo treatment. While EPR3776 staining, which binds to the C-terminus of vimentin (aa 425–466), was mainly co-localized with 84-1 on control cells, the CSV-specific staining was predominantly found in the absence of EPR3776, and particularly in Doxo-treated cells, implying the presence of a modified vimentin structure, which could not be detected by EPR3776. Further information about the CSV-specific antibody can be found in the patent (international publication number WO2014138183A1). This preliminary experiment implies that the structure of externalized vimentin is possibly modified. Moreover, we performed a scanning electron microscopy (SEM) analysis, which indicated that CSV does not form a filamentous structure (*Figure 2i*). That CSV might form oligomers of about 4–12 monomers instead of filaments has been previously been proposed (*Hwang and Ise, 2020*).

Immunofluorescence staining of intracellular vimentin revealed that Doxo-treated chondrocytes exhibited an abnormal vimentin arrangement, characterized by increased vimentin bundling, or a (nearly) complete loss of vimentin (*Figure 2j*). Correlation analysis between CSV-positivity and relative mRNA levels of *VIM* confirmed a negative association between the gene expression and translocation of vimentin (*Figure 2k*). These findings indicate that the disruption of the vimentin network might be connected to a SIPS-like phenotype and CSV expression.

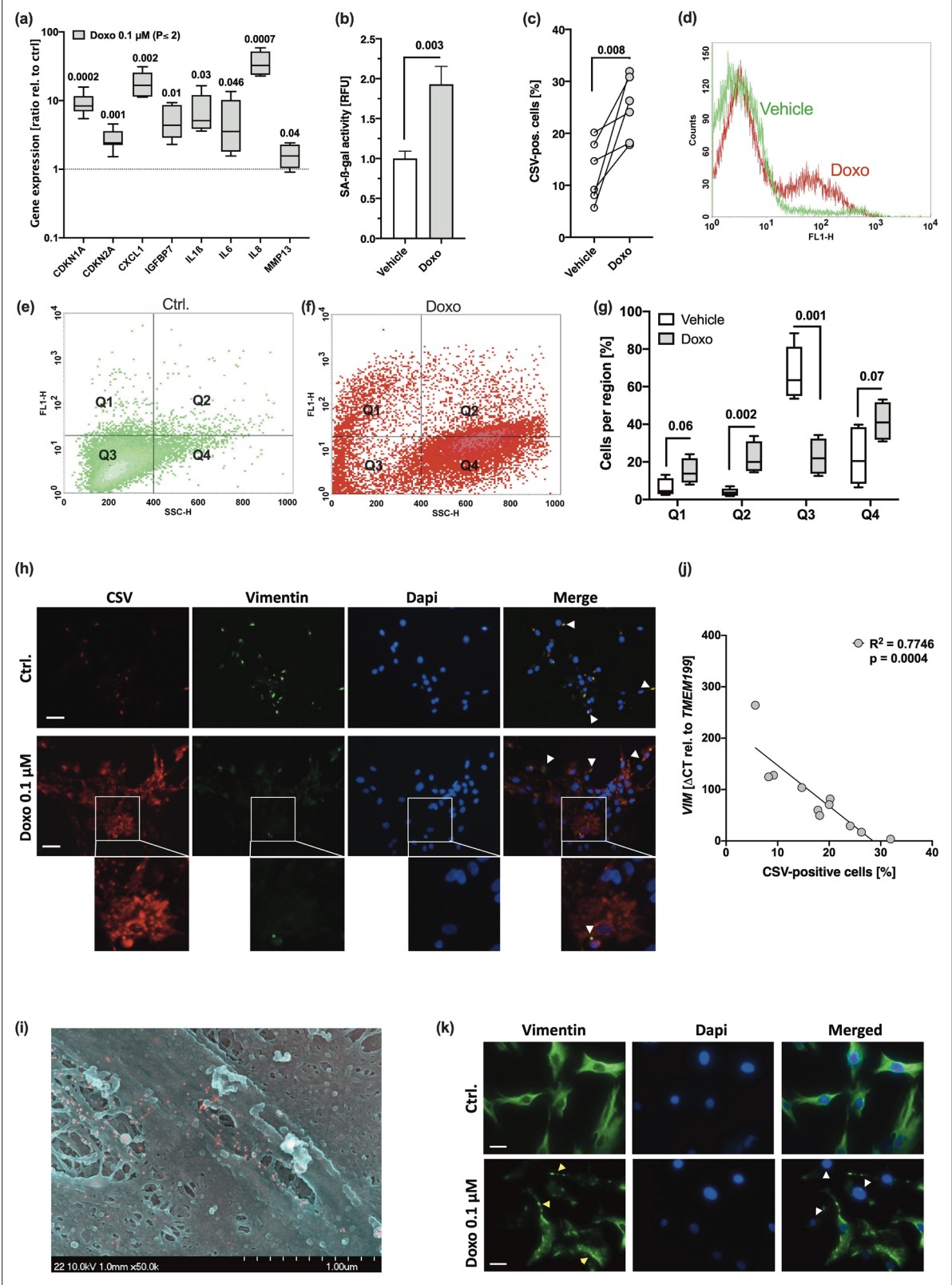

**Figure 2.** Doxorubicine (Doxo) stimulation leads to stress-induced premature senescence (SIPS) and enhanced cell surface vimentin (CSV)-positivity in human articular chondrocyte (hAC). hAC were stimulated with 0.1 µM Doxo for 10 days. (**a**) mRNA levels of senescence-associated (SA) markers, including *CDKN1A, CDKN2A, CXCL1, IGFBP7, IL1ß, IL6, IL8,* and *MMP13* (n=5). (**b**) Activity of senescence-associated-β-galactosidase (SA-β-gal) in untreated and Doxo-stimulated hAC (n=5). (**c**) Presence of CSV was determined by means of cytometric analysis; (**d**) exemplary histograms of untreated

*Figure 2 continued on next page*

Figure 2 continued

(green) and Doxo-stimulated (red) hAC. Exemplary distribution of (**e**) untreated (green) and (**f**) Doxo-stimulated (red) hAC during cytometric analysis and (**g**) corresponding statistical analysis (n=4). (**h**) Immunofluorescence staining of CSV [84-1] and vimentin [EPR3776] on the surface of unpermeabilized untreated and Doxo-stimulated hAC; white arrow tips indicate double-positive regions (appearing yellow); scale bar: 50 µm. (**i**) Exemplary scanning electron microscopy (SEM) image of Doxo-treated hAC; positive immunostaining of CSV appears in red. (**j**) Correlation analysis of *VIM* mRNA levels and CSV-positivity in hAC from the Doxo experiment, including untreated and Doxo-treated cells. (**k**) Immunofluorescence staining of the intracellular vimentin network in hAC 10 days after Doxo stimulation; yellow arrow tips indicate vimentin bundles; white arrow tips indicate (nearly) complete loss of vimentin; scale bar: 25 µm. Data are presented as scatter plot; box plots, median, whiskers min to max; or column bars with mean, SEM. Statistics: (**a**) and (**g**) were analyzed by a multiple t test; (**b**) and (**c**) were analyzed by means of paired two-tailed t test; (**j**) was analyzed by a Pearson correlation analysis. Each data point represents an independent biological replicate (**n**).

## Knockdown of *CDKN2A* cannot completely reverse but attenuate CSV and SA marker expression

The *CDKN2A* gene encodes for p14ARF and p16INK4A, which are crucial regulators in cell cycle arrest and consequent senescence. As described above, both CSV and CDKN2A expression were similarly enhanced in OA as well as after cartilage trauma or Doxo stimulation. Therefore, we assumed a connection between *CDKN2A*-mediated senescence and CSV-positivity.

As siRNA-mediated silencing of *CDKN2A* was previously described to result in a recovery of chondrogenic characteristics and reduction of SA-β-gal in OA hAC (*Zhou et al., 2004*), we considered a knockdown of *CDKN2A* to further clarify the impact of p14ARF/p16INK4A on chondrosenescence and CSV-positivity. By means of a special transfection regime (*Figure 3a*), we achieved a knockdown of *CDKN2A* in hAC over 7 days, which was confirmed by qRT-PCR and immunofluorescence staining (*Figure 3b–d*). Moreover, knockdown of *CDKN2A* significantly decreased the gene expression of SA markers *CDKN1A*, *CXCL1*, and *IL8* (*Figure 3e*) and enhanced cell proliferation (*Figure 3f*). However, reduction in *CDKN2A* could not completely reverse the senescent phenotype and had only minor, though significant, effect on SA-β-gal activity (*Figure 3g and h*) and CSV (*Figure 3i*). Subsequent correlation analysis between relative *CDKN2A* expression and CSV-positivity did not reveal any significant association in case of the knockdown experiment but after Doxo stimulation (*Figure 3j*). Taken together, knockdown of *CDKN2A* could not completely reverse CSV presentation on hAC, implying that externalization of vimentin and chondrosenescence do not exclusively depend on CDKN2A.

## Knockdown of VIM results in enhanced CSV-positivity and a SIPS-like phenotype in hAC

Overexpression of vimentin was found to induce a senescence-like morphology in fibroblasts, characterized by a flattened and enlarged cell shape (*Nishio et al., 2001*). However, disruption of the vimentin network has been associated with a decline in chondrogenic characteristics and an OA phenotype in chondrocytes (*Blain et al., 2006*). As we assumed a connection between alterations in the intracellular vimentin network and CSV, we performed an siRNA-mediated knockdown of *VIM* in hAC.

By using the transfection regime depicted before (*Figure 3a*), we applied a knockdown over 7 days, which was confirmed by quantitative real-time PCR (qRT-PCR), flow cytometry, and immuno-fluorescence staining (*Figure 4a–c*). Reduction in intracellular vimentin did not result in cell death. However, we observed that the typical cage-like distribution of vimentin around the nuclei was lost and the filaments were condensed in *siVIM*-transfected hAC, indicating a disruption of the remaining vimentin network (*Figure 4c*). Furthermore, silencing of *VIM* resulted in a progressive alteration of the cell morphology, comparable to that of OA hAC (*Supplementary file 1*) and provoked significant accumulation of ROS and SA-β-gal (*Figure 4d–g*). In line with this, the gene expression levels of various SA markers were enhanced, which was significant in case of *CDKN1A*, *CDKN2A*, and *IL6* (*Figure 4h*). Additionally, we confirmed nuclear accumulation of p53 (*Figure 4i*) and found indications of cell cycle arrest in *siVIM*-transfected hAC as demonstrated by lower proliferation and an increased percentage of cells in $G_0/G_1$ phase (*Figure 4j and k*). Moreover, silencing of *VIM* clearly suppressed COL2 production of hAC (*Supplementary file 1*). These findings are in agreement with *Tolstonog et al., 2001*, who observed enhanced oxidative stress and SIPS in embryonic fibroblasts from vimentin knockout mice.

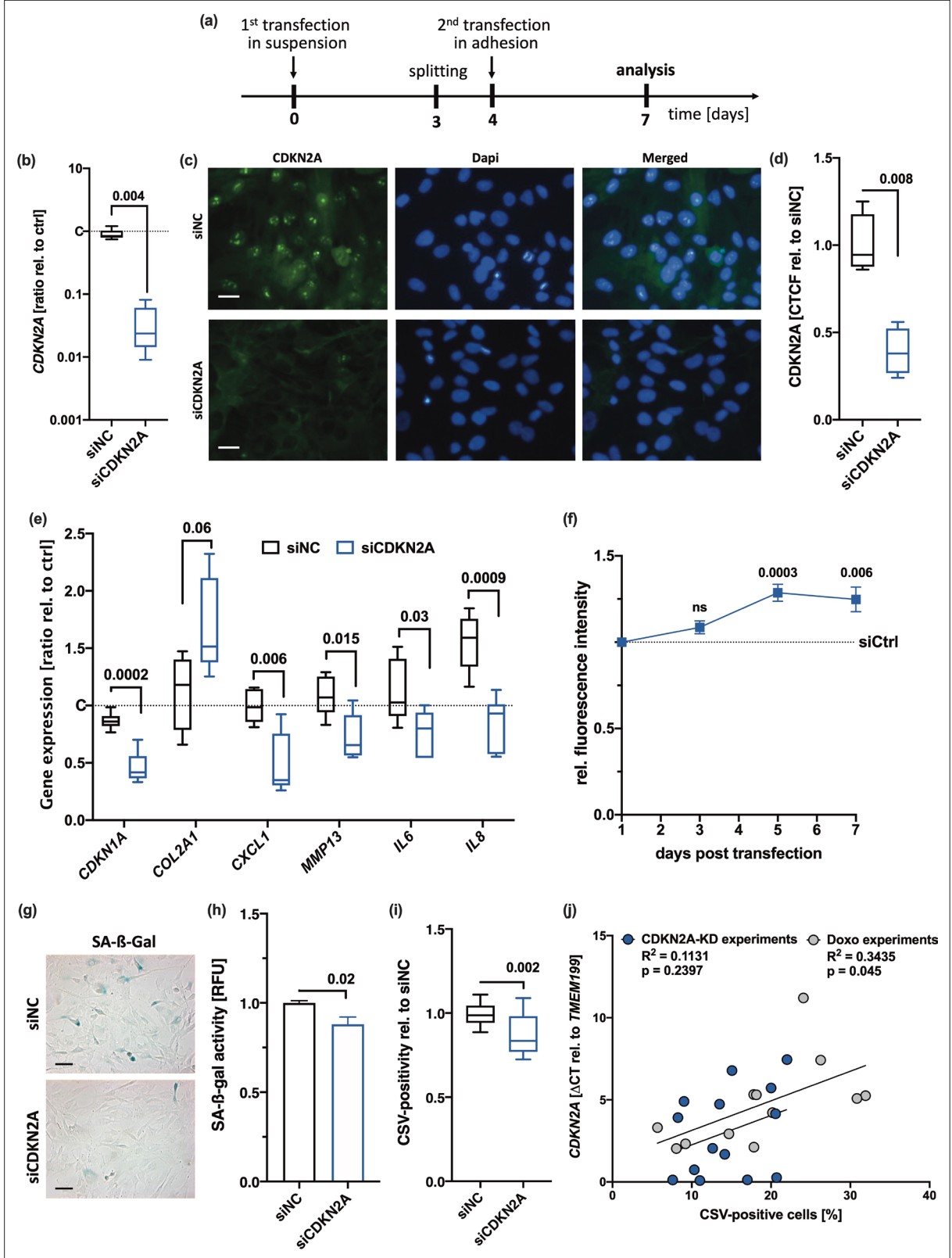

**Figure 3.** Influence of CDKN2A knockdown on senescence-associated (SA) markers and cell surface vimentin (CSV). (**a**) Outline of the transfection regime. (**b–d**) siRNA-mediated knockdown of CDKN2A was confirmed by means of (**b**) quantitative real-time PCR (qRT-PCR) (n=8) and (**c**) by means of immunofluorescence; scale bar: 25 µm, (**d**) including quantification of the corrected total cell fluorescence (CTCF) values relative to siNC (n=4). (**e**) mRNA levels of SA markers, including *CDKN1A, CXCL1, MMP13, IL6,* and *IL8* normalized on untransfected controls (n≥5). (**f**) Mitotic activity and proliferation,

*Figure 3 continued on next page*

*Figure 3 continued*

respectively, was determined by means of an alamarBlue assay performed at different time points (3 days, 5 days, 7 days). (**g**) Exemplary senescence-associated-β-galactosidase (SA-β-gal) staining and (**h**) corresponding quantification of the SA-β-gal activity in human articular chondrocyte (hAC) after transfection (n=6); scale bar: 50 µm. (**i**) CSV-positivity of hAC was assessed by means of cytometric analysis (n=8). (**j**) Correlation analysis of *CDKN2A* mRNA levels and CSV-positivity in hAC from different experiments. Data are presented as box plots with median, whiskers min to max; or column bars with mean, scanning electron microscopy (SEM). Statistics: (**b**), (**d**), (**h**), and (**i**) were analyzed by a paired two-tailed t test; (**e**) and (**f**) were analyzed by means of a multiple t test; (**j**) was analyzed by a Pearson correlation analysis. Each data point represents an independent biological replicate (**n**).

Interestingly, CSV was significantly enhanced after knockdown of *VIM* (*Figure 4l and m*). Correlation analysis between CSV-positivity and relative mRNA levels of *VIM* confirmed a negative association between the gene expression and translocation of vimentin (*Figure 4n*).

Therefore, we assumed that chondrosenescence and consequent increase of CSV can be attributed to the disturbance of the intracellular vimentin network and subsequent oxidative stress.

## Collapse of the vimentin network is associated with enhanced CSV-positivity and a SIPS-like phenotype in hAC

To further confirm our assumption that disturbance of the vimentin network results in chondrocyte senescence and vimentin dislocation to the membrane, we investigated if direct disruption of the vimentin network by simvastatin (Sim) might increase CSV and promote SIPS. According to *Trogden et al., 2018*, who previously described the effects of Sim on the intermediate filament in adrenal carcinoma SW13 cells, we observed a reorganization of intracellular vimentin. Co-staining with phalloidin confirmed that Sim had only minor effect on the actin cytoskeleton but resulted in a strong condensation of vimentin filaments, which were selectively located in close proximity to the nuclei (*Figure 5a*). Similar to the phenotype observed after siRNA-mediated *VIM* knockdown, the vimentin 'cage' around the nucleus was collapsed, resulting in smaller size and elliptical shape of the nuclei (*Figure 5a*). Sim-treated hAC further exhibited enhanced levels of CSV (*Figure 5b and c*), cytoplasmatic and mitochondrial ROS (*Figure 5d–f*), and SA-β-gal activity (*Figure 5g, h*). Accumulation of mitochondrial ROS might be explained by the fact that some cytoskeletal components, and in particular vimentin, play a crucial role in mitochondrial morphology and function (*Tolstonog et al., 2001*; *Mado et al., 2019*).

Overall, these results support the assumption that CSV on hAC is triggered by breakdown or structural alteration of the intracellular vimentin network and subsequent cellular stress and senescence.

## CSV is associated with chondrocyte plasticity and osteogenic differentiation in vitro

Chondrocytes are known to rapidly lose their chondrogenic phenotype and undergo SA alterations in 2D culture (*Lee et al., 2019*). Dedifferentiated chondrocytes at passages ≥4 are characterized by enhanced levels of stem cell-associated markers (*Fickert et al., 2004*; *Barbero et al., 2003*), augmented cellular plasticity, and tri-linear differentiation capacities (*Barbero et al., 2003*; *Caron et al., 2012*; *Varela-Eirín et al., 2018*). Interestingly, Varela-Eirín et al. described increasing levels of CD105 and CD166 during in vitro expansion of chondrocytes as indicator of an immature phenotype with high cellular plasticity but also SA characteristics (*Varela-Eirín et al., 2018*).

Consequently, we evaluated CSV levels on hAC during in vitro expansion and its association to cellular plasticity. In fact, CSV increased in a passage-dependent manner (*Figure 6a and b*) and strongly correlated with the expression of CD105 and CD166 (*Figure 6c*).

To further investigate the role of CSV as a potential marker of cellular plasticity in hAC, dedifferentiated chondrocytes (passages 4–5) were re-differentiated into osteogenic or chondrogenic lineage for 28 days, which was confirmed by expression analysis of corresponding markers and exemplary staining (*Supplementary file 1*). CSV levels were significantly diminished after differentiation, while subsequent culturing at basal conditions and sub-confluent density restored CSV (*Figure 6d*) and decreased osteogenic as well as chondrogenic markers (*Supplementary file 1*-Figure S3a,b,d,e). Additional mRNA analysis confirmed the dynamics of CD105 and CD166 as well as VIM expression during the differentiation experiment, and lower mRNA levels of VIM in dedifferentiated chondrocytes (*Supplementary file 1*-Figure S4a-c), reinforcing the assumption that VIM gene expression is inversive to the presentation of CSV.

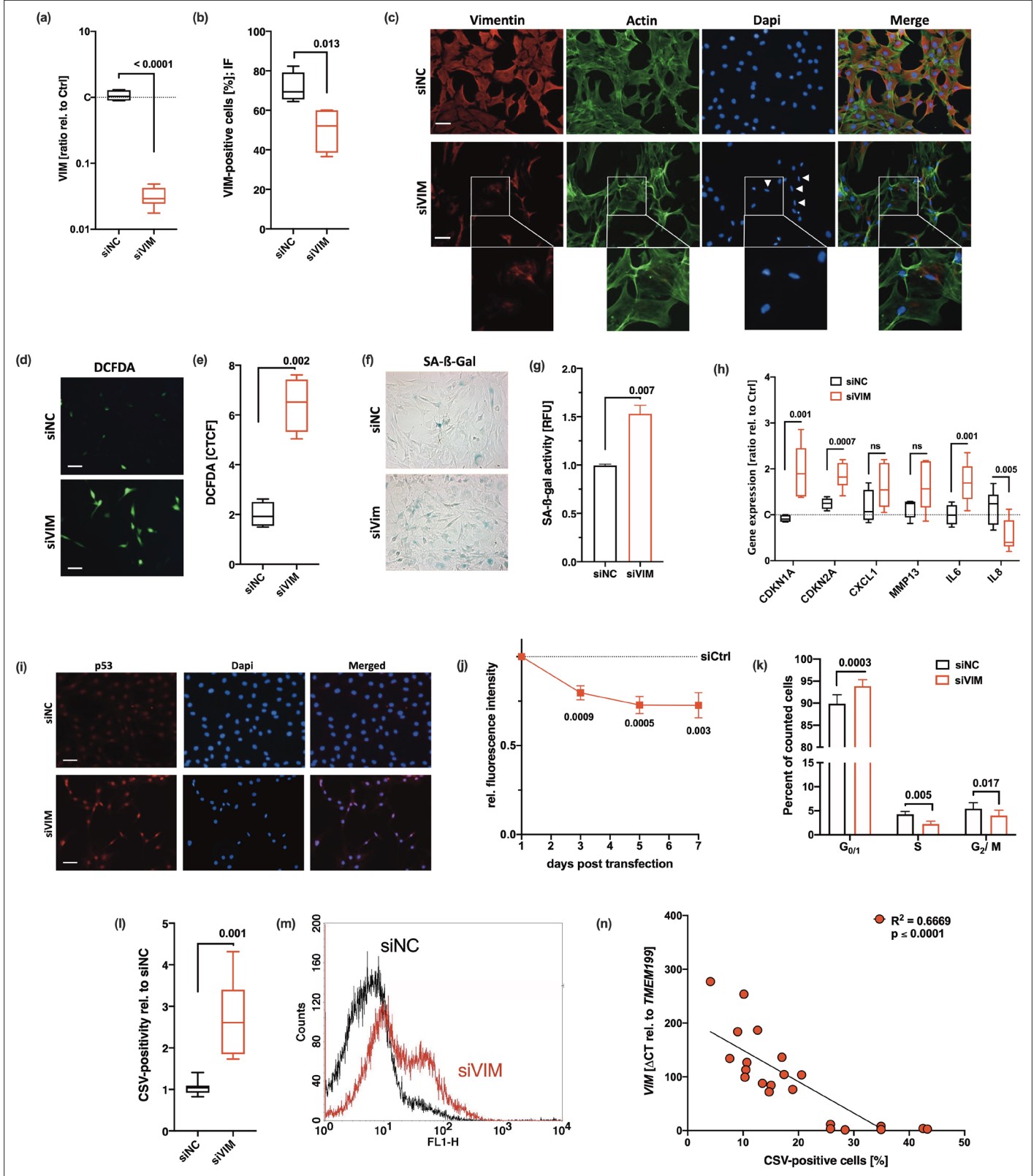

**Figure 4.** Cellular consequences of siRNA-mediated knockdown of vimentin in human articular chondrocyte (hAC). (**a–c**) siRNA-mediated knockdown of VIM was confirmed by means of (**a**) quantitative real-time PCR (qRT-PCR) (n=8), (**b**) flow cytometry (n=4), and (**d**) exemplary co-staining of vimentin and actin cytoskeleton; white arrow tips indicate aberrant nuclei (smaller size and elliptical shape); scale bar: 50 μm. (**d**) Exemplary images of DCFDA assay and (**e**) corresponding quantification of the corrected total cell fluorescence (CTCF); scale bar: 50 μm. Senescent phenotype of hAC after knockdown

*Figure 4 continued on next page*

*Figure 4 continued*

of VIM was determined by (**f**) exemplary senescence-associated-β-galactosidase (SA-β-gal) staining and (**g**) quantification of the SA-β-gal activity, (**h**) mRNA levels of SA markers, including *CDKN1A*, *CDKN2A*, *CXCL1*, *MMP13*, *IL6*, and *IL8* (all n≥5) and (**i**) exemplary staining of p53. Mitotic activity and proliferation, respectively, were determined by means of (**j**) an alamarBlue assay performed at different time points (3 days, 5 days, 7 days; n≥5) as well as (**k**) a flow cytometry-based cell cycle assay. (**l**) Flow cytometric analysis of cell surface vimentin (CSV)-positivity on transfected hAC; (**m**) exemplary histograms of hAC transfected with siNC (black) or siVIM (red); (n≥5). (**n**) Correlation analysis of *VIM* mRNA levels and CSV-positivity in hAC from the transfection experiment, including untransfected, *siNC*, and *siVim* cells. Data are presented as box plots with median, whiskers min to max; column bars with mean and scanning electron microscopy (SEM); or points and connecting line with mean and SEM. Statistics: (**a**), (**b**), (**e**), (**g**), and (**l**) were analyzed by a paired two-tailed t test; (**h**), (**j**), and (**k**) were analyzed by means of a multiple t test; (**n**) was analyzed by a Pearson correlation analysis. Each data point represents an independent biological replicate (**n**). Each data point represents an independent biological replicate (**n**). Abbreviations: siNC = cells transfected with scrambled siRNA; siVim = cells transfected with vimentin-targeting siRNA.

Since CSV correlated with stem-like markers, we wondered whether CSV might be a progenitor-specific marker. While isolated bm-MSC, indeed, exhibited very high levels of CSV, we could not find any significant difference in CSV levels between passage-matched hAC and cartilage-derived chondrogenic stem progenitor cells (CSPC) (***Figure 6e***). Due to the fact that CSV was linked to a dysfunctional behavior in hAC, so far, we assumed that CSV might indicate a compromised chondrogenic potential. Indeed, we found a positive correlation between CSV-positivity and matrix calcification after osteogenic differentiation of high-passage hAC (***Figure 6f***), while in vitro chondrogenesis was declined in a CSV-dependent manner (***Figure 6g***).

Altogether these findings suggest that an increased level of CSV on hAC is associated with an enhanced cellular plasticity but decreased chondrogenic capacities.

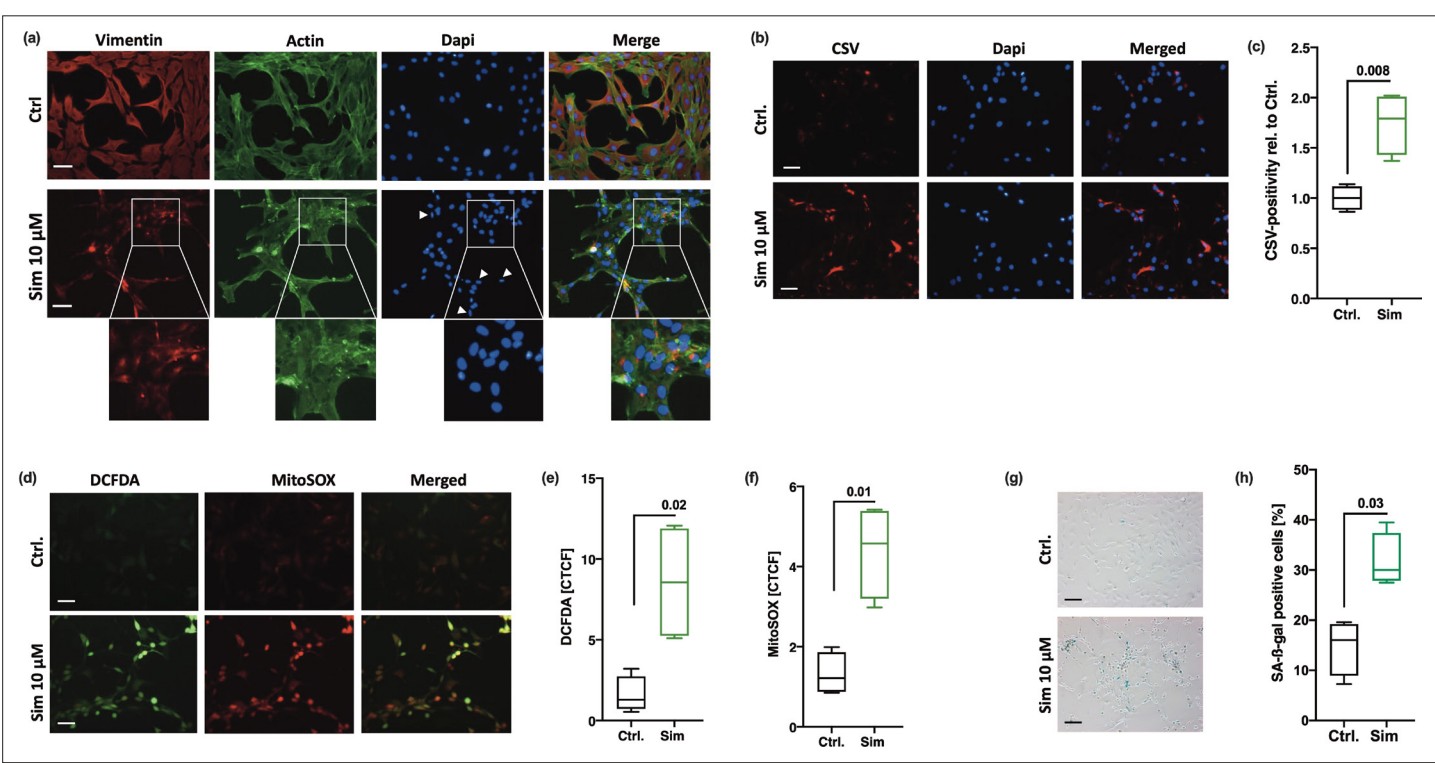

**Figure 5.** Consequences of simvastatin (Sim)-mediated collapse of the vimentin network in human articular chondrocyte (hAC). (**a**) Immunofluorescence staining of the intracellular vimentin and actin network in hAC 20 hr after Sim treatment (10 µM). (**b**) Assessment of cell surface vimentin (CSV) on hAC treated with Sim by means of immunofluorescence staining and (**c**) corresponding quantification of CSV-positivity using flow cytometry; white arrow tips indicate aberrant nuclei (smaller size and elliptical shape); scale bar: 50 µm. (**d**) Fluorescence-based analysis of cytoplasmatic (DCFDA) and mitochondrial (MitoSOX) reactive oxygen species (ROS) in hAC; (e+f) including corresponding quantification of the corrected total cell fluorescence (CTCF). (**g**) Senescence-associated-β-galactosidase (SA-β-gal) staining and (**h**) activity assay in Sim-treated hAC. Data are presented as box plots with median, whiskers min to max; n=4 each. Statistics: (**c**), (**e**), (**f**), and (**h**) were analyzed by a paired two-tailed t test. Each data point represents an independent biological replicate (**n**).

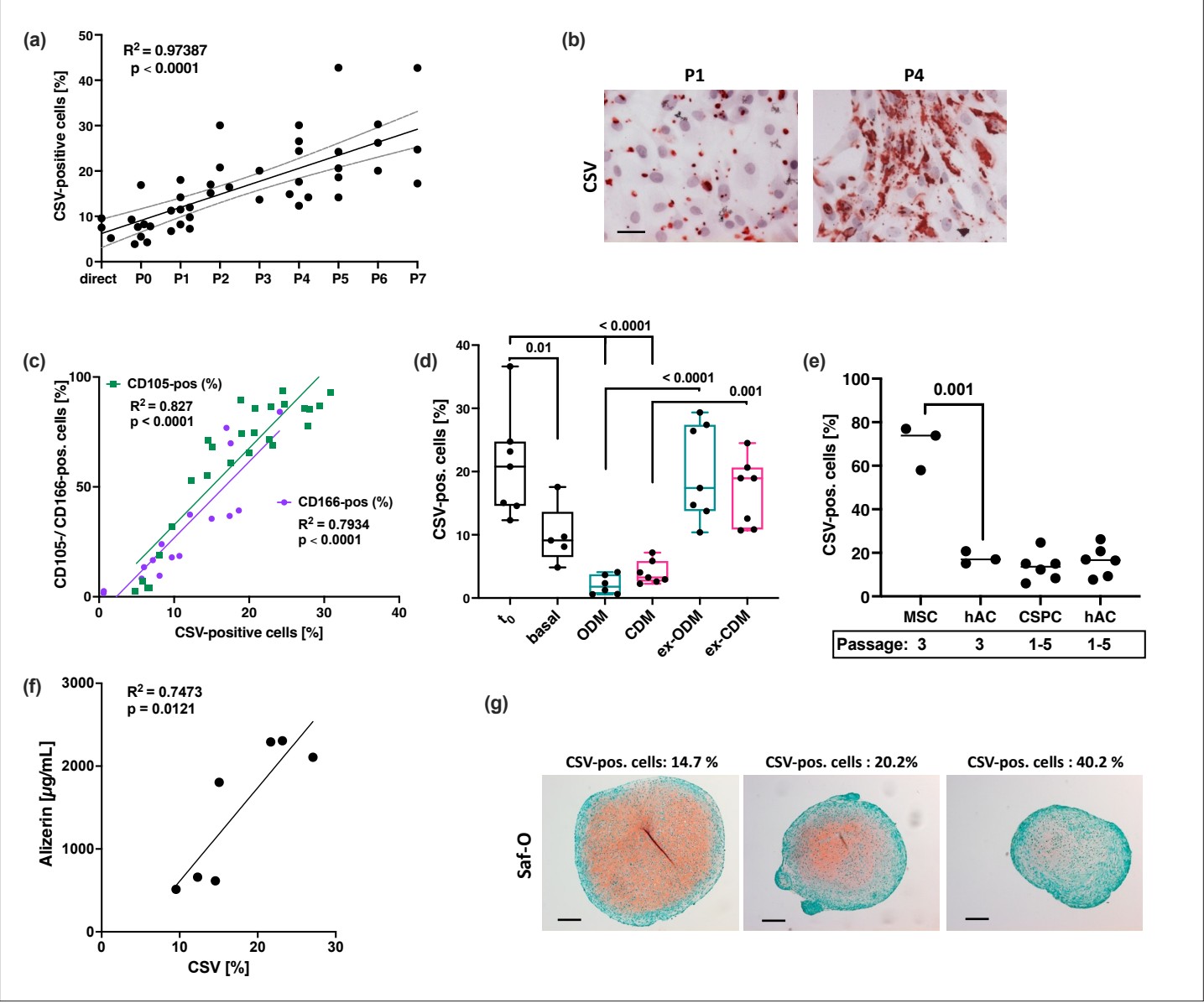

**Figure 6.** Association between cell surface vimentin (CSV) and chondrocyte plasticity. (**a**) Correlation analysis between passage number and percentage of CSV-positive cells (determined by means of flow cytometry). Analysis includes n=18 different donors, some donors were measured at various passages. (**b**) Exemplary images of immunocytochemistry staining against CSV on human articular chondrocyte (hAC) at low (**P1**) and high (**P4**) passage. (**c**) Correlation analysis between percentage of CSV-positive and CD105 (green; n=26) or CD166-positive (purple; n=16) hAC; surface markers were determined by means of flow cytometry. (**d**) hAC were cultured in basal, osteogenic (ODM; turquoise), or chondrogenic (CDM; pink) medium for 28 days. Afterward, one half of the cells were further cultured for 14 days in basal medium at sub-confluence (=ex-ODM and ex-CDM). Percentage of CSV-positive cells was determined via flow cytometry before differentiation ($t_0$), directly after differentiation at day 28 (basal, ODM, CDM), or after further expansion at basal conditions at day 42 (ex-ODM, ex-CDM); n≥5. (**e**) Percentage of CSV-positive cells in isolated hAC (n=3 and n=6), bm-MSC (n=3), and chondrogenic stem progenitor cell (CSPC) (n=6) was determined by flow cytometry; bm-MSC and CSPC, respectively, were compared to passage-matching hAC. (**f**) Matrix calcification ($Ca^{2+}$ deposition) was quantified via Alizarin red staining 28 days after osteogenic differentiation of hAC (n=7). Alizarin red concentrations were correlated with the percentage of CSV-positive cells in the corresponding donor, which was determined by flow cytometry. (**g**) Exemplary images of safranin-O-stained hAC pellets after chondrogenic differentiation for 28 days; images of three independent donors with increasing CSV-positivity (from left to right) are shown. Glycosaminoglycans appear in red and imply production of hyaline cartilage. Size of the pellet is considered as indicator of matrix production and successful chondrogenesis. Statistics: (**a,c,f**) were analyzed by a Pearson correlation analysis; (**d,e**) were analyzed by means of a one-way ANOVA, including a Sidak post hoc test.

## High CSV levels are associated with enhanced plastic adherence

Vimentin is known to contribute to integrin-mediated cell adhesion and binding to ECM components, respectively (*Bhattacharya et al., 2009*). Interestingly, we observed that the adhesion of hAC on culture plates increased with higher CSV levels. Accordingly, cells detached within 20 min upon EDTA exposure had significantly lower CSV levels as compared to the cells, which remained on the plates (*Figure 7a*). By means of an adherence assay, it could be confirmed that donors with low CSV-positivity (below 14%) detached significantly faster as compared to that with CSV levels of 16% and higher (*Figure 7b*). Subsequent correlation analysis revealed a significant association between the CSV-positivity and adherence (*Figure 7c*). A similar outcome was obtained by using the cancer cell lines HeLa (low CSV) and SaOs-II (high CSV) (*Supplementary file 1*). Further investigation regarding the adhesion of hAC to different ECM components demonstrated a significant association between CSV levels and binding affinity to fibronectin and vitronectin, while no association was found in case of tenascin (*Figure 7d*) or Collagen I, II, and IV (*Figure 7e*) as determined by means of an ECM cell adhesion array. Interestingly, knockdown of vimentin, which induced cellular stress and enhanced CSV levels as described above, significantly increased the gene expression of αV, α5, and β1 integrin subunits in hAC (*Figure 7f*).

In sum, these results imply that CSV contributes to adhesion strength of hAC on plastic and might be involved in the interaction with certain ECM components.

## Discussion

Senescent chondrocytes are characterized by a dysfunctional behavior, which is thought to contribute to ongoing cartilage degeneration and thus OA progression (*Jeon et al., 2018*; *Coryell et al., 2021*). Here, we provide first evidence for the importance of the vimentin network in chondrosenescence. In this context, we introduce CSV as a membrane-associated indicator of vimentin disruption and consequent damage and senescence in hAC. In sum, we could demonstrate that CSV was (1) enhanced in damaged and dysfunctional hAC after traumatic injury or due to OA and (2) was associated with an OA-like phenotype as well as cell plasticity. Moreover, CSV on hAC was (3) strongly enhanced in response to the disruption of the intracellular vimentin network and consequent stress and (4) loss of the chondrogenic capacities (*Figure 7g*).

Vimentin is an essential component of the cytoskeleton. Besides cell morphology and mechanical integrity, vimentin is involved in different cellular processes, including migration, adhesion, and differentiation (*Bobick et al., 2010*; *Ivaska et al., 2007*). However, overexpression of vimentin is also considered as a hallmark of EMT in cancer cells (*Ivaska et al., 2007*) and was found to induce a senescent cell morphology in fibroblasts (*Nishio et al., 2001*). In chondrocytes, IL-1β stimulation resulted in reduced gene expression of vimentin via p38-signaling, which is also associated with posttraumatic cartilage degeneration and OA progression (*Joos et al., 2008*; *Ding et al., 2010*). Moreover, disassembly and collapse of the cage-like vimentin network, observed after acrylamide treatment or traumatic cartilage loading, has been linked to decreased cell stiffness as found in OA chondrocytes (*Henson and Vincent, 2008*; *Haudenschild et al., 2011*). Accordingly, it has been hypothesized that the vimentin network is involved in the maintenance of the chondrogenic phenotype while its disassembly, in turn, can disturb chondrocyte characteristics, contributing to the development of OA (*Blain et al., 2006*). Beyond that, we provide clear evidence that alterations of the vimentin network and enhanced CSV levels, respectively, are associated with compromised chondrogenic capacities and a SIPS phenotype.

Besides increased CSV levels, Sim-mediated breakdown of the vimentin network remarkably changed the nuclear shape and enhanced mitochondrial ROS levels. In fact, vimentin filaments are involved in the regulation of the nuclear shape, mechanics, and chromatin condensation, as well as mitochondrial structure and function (*Trogden et al., 2018*; *Keeling et al., 2017*). The interaction between vimentin and mitochondria depends on the small GTPase Rac1 – a target of Sim (*Matveeva et al., 2015*). In contrast to other vimentin-binding molecules, such as withaferin A, which also affect microtubules and microfilaments, Sim is considered to bind vimentin with high specificity (*Trogden et al., 2018*; *Grin et al., 2012*; *Lavenus et al., 2020*). Although enhanced oxidative stress and subsequent induction of senescence via the p53/p21 pathway has been previously found in Sim-treated

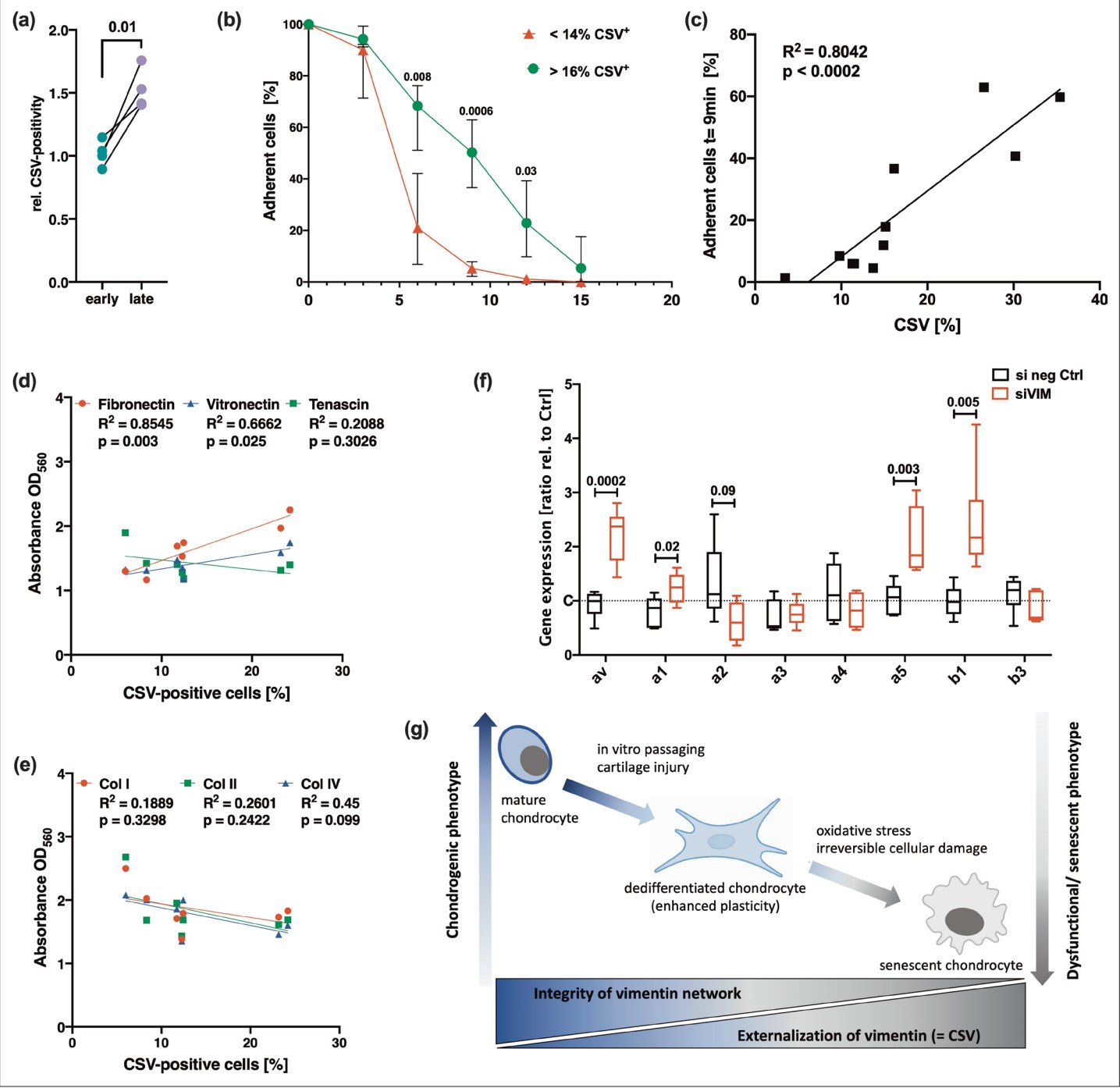

**Figure 7.** Potential role of cell surface vimentin (CSV) in cell adhesion and outline of hypothesis. (**a**) Relative amount of CSV-positivity in human articular chondrocyte (hAC) detached after 20 min of EDTA exposure (referred to as 'early detached cells'=early) and hAC from the same culture plate, which required mechanically detachment using a cell scraper (referred to as 'late detached cells'=late); n=4. (**b**) Adhesion strength in hAC with a relatively low percentage of CSV-positive cells (<14%; range: 3.5–13.7%) was compared to hAC with a relatively high percentage of CSV-positive cells (>16%; range: 16.2–35.4%) by means of a cell adhesion assay (each group n=4). (**c**) Correlation analysis between the percentage of CSV-positive cells and the percentage of remaining (adherent) cells on the culture plate after exposure to EDTA for 9 min of the corresponding donor (n=11). (**d,e**) Correlation analysis between the percentage of CSV-positive cells and the adherence capacity to different extracellular matrix (ECM) components of the corresponding donor (n=7); attachment of cells on different substrates was determined by means of a commercial ECM cell adhesion array and given as 'absorbance at OD560'. (**f**) mRNA levels of integrin subunits, including *ITGAV* (αV), *ITGA1* (α1), *ITGA2* (α2), *ITGA3* (α3), *ITGA4* (α4), *ITGA5* (α5), *ITGB1* (β1), and *ITGB2* (β2) (all n≥5). (**g**) Outline of our hypothesis: Loss of the chondrogenic phenotype is associated with alterations of the intracellular vimentin network and consequent expression of CSV. While mild stress or in vitro passaging promotes chondrocyte dedifferentiation, excessive stress

*Figure 7 continued on next page*

*Figure 7 continued*

and irreversible damage eventually results in cellular senescence and enhanced CSV levels. Statistics: (**a**) was analyzed by a paired two-tailed t test; (**b,f**) was analyzed by means of a multiple t test; (**c,d,e**) were analyzed by a Pearson correlation analysis.

human melanoma cells (*Guterres et al., 2013*), it should be noted that Sim has also been described as an antioxidative and chondroprotective drug in cartilage (*Riegger et al., 2022*).

Previously, other surface-associated markers have been reported on senescent chondrocytes, such as urokinase plasminogen activator surface receptor (uPAR) (*Kirsch et al., 2022*; *Schwab et al., 2004*), dipeptidyl peptidase 4 (DPP-4) (*Bi et al., 2019*), and connexin43 (*Varela-Eirín et al., 2018*). Except for DPP-4, all three surface markers (CSV, connexin43, and uPAR) are connected with EMT, thus indicating alteration in cell properties, in particular migration and adhesion. Moreover, increased expression of different EMT makers, such as Twist-1, N-cadherin, and vimentin, have been reported in OA chondrocytes (*Varela-Eirín et al., 2018*; *Hasei et al., 2017*). Nevertheless, the hypothesis that chondrocytes might undergo an EMT-like process remains controversially discussed, because chondrocytes are mesenchymal and not epithelial cells. In a recent review, *Gems and Kern, 2022* propose to consider senescent chondrocytes as activated and hyperfunctional remodeling cells occurring during OA progression (*Loeser, 2014*). Accordingly, chondrosenescence might represent an unsuccessful attempt of tissue repair. They further suppose that the senescent or activated chondrocytes are associated with a hypertrophic, bone-forming phenotype, following the process of bone development rather than hyaline cartilage formation. In line with this, we observed that CSV was associated with enhanced osteogenic capacities and a decline in chondrogenic properties.

Moreover, CSV has also been linked to stemness, due to its strong expression on stem cell-like cancer cells as well as on rat bmMSC (*Mitra et al., 2015*; *Ise et al., 2019*). However, neither the underlying mechanism of vimentin externalization nor its biological function could be sufficiently clarified so far. Previously, *Frescas et al., 2017* identified an MDA-modified vimentin, on the cell surface of senescent fibroblasts. They hypothesized that the presentation of MDA-vimentin might serve as a 'eat me'-signal to phagocytes. Although CSV has also been described on apoptotic neutrophils, it could not be clarified whether this really represents an 'eat me'-signal or not (*Moisan and Girard, 2006*). In our study, we could not observe any connection between cell death and CSV-positivity of hAC, not even after direct induction of apoptosis by means of cycloheximide and TNF (*Supplementary file 1*).

In the current study, we observed that CSV on chondrocytes was reduced by siRNA-mediated silencing of CDKN2A and increased after Doxo treatment or cartilage trauma. While we confirmed that mRNA levels of both CDKN1A and CDKN2A were significantly enhanced upon injury but exhibited different expression levels over time, we determined CSV-positive cells only at one time point after ex vivo cartilage trauma. Future studies might also consider earlier and later time points after cartilage injury to identify a potential time-dependent peak or decline in CSV-positive chondrocytes. In this way a potential association between CSV and the expression levels of CDKN1A and CDKN2A, which are thought to play differential roles in initiating and maintenance of senescence, respectively (*Riegger et al., 2023*), might be clarified.

Our preliminary results concerning the biological function of CSV imply that CSV promotes plastic adhesion, as well as the binding affinity to fibronectin and vitronectin. Both observations might be closely linked as plastic adhesion is mainly mediated by fetal bovine serum (FBS)-derived vitronectin and/or fibronectin. In chondrocytes, interaction with fibronectin and vitronectin mainly involves $\alpha5\beta1$ and $\alpha V\beta3$ integrins, which both recognize the Arg-Gly-Asp (RGD) sequence of these ECM components (*Loeser, 2014*). Interestingly, it has previously been described that $\beta1$ and $\beta3$ integrins can recruit vimentin to the cell membrane, where the intermediate filaments interact with focal adhesion proteins and increase the adhesive strength on plastic (*Bhattacharya et al., 2009*). Moreover, intracellular vimentin was found to recruit paxillin to the focal adhesion complex, which enhanced $\beta1$ integrin clustering and consequent collagen binding (*Ostrowska-Podhorodecka et al., 2021*). Although we found clear indications for an association between CSV and cell adhesion, as well as increased mRNA levels of certain integrin subunits after silencing of vimentin, the underlying mechanisms and potential interdependence with integrins clearly deserve further investigation.

The usage of human material should be considered as a strength of this study; however, the OA origin also represents a crucial limitation. Although the material was classified in macroscopically intact (grade $\leq1$) and highly degenerated (grade $\geq3$) tissue, it cannot be completely excluded that the

intact cartilage has been affected by OA on molecular level. Nevertheless, we still could demonstrate significant differences regarding SA marker expression in highly degenerated or impacted cartilage as compared to macroscopically intact tissue.

In sum, our study provides first evidence of CSV as a novel surface marker of phenotypical alteration and SIPS in hAC. Although we could not clarify the underlying mechanism of vimentin translocation, the negative association between intracellular vimentin levels and CSV-positivity implies that the externalization does not require additional vimentin but rather represents a reallocation of existing filaments in response to cellular stress. Overall, our findings suggest that presentation of CSV most likely results from vimentin network disturbance and is associated with a dysfunctional phenotype in hAC.

## Materials and methods

### Isolation and cultivation of hAC and CSPC

hAC were enzymatically isolated from human cartilage with either OARSI grade ≥3 (highly degenerated) or OARSI grade ≤1 (macroscopically intact) (*Waldstein et al., 2016*). In short, full-thickness cartilage was minced and digested for 45 min with 0.2% pronase (Sigma-Aldrich) and overnight with 0.025% collagenase (Sigma-Aldrich). CSPC were isolated by outgrowth of the migratory active cells from intact human cartilage by as previously described (*Riegger et al., 2018b*).

Cells were cultured in serum-containing chondrocyte (=basal) medium (1:1 DMEM 1 g/L glucose/Ham's F12, 10% FBS, 0.5% penicillin/streptomycin, 2 mM L-glutamine, and 10 µg/mL 2-phospho-L-ascorbic acid trisodium salt) and split at a confluence of 80%.

### Cartilage explant preparation, cultivation, and impact loading

Full-thickness cartilage explants (Ø=6 mm) from macroscopically intact tissue (OARSI score ≤1) were harvested and cultivated in chondrocyte medium for 24 hr in an incubator (37°C, 5% $CO_2$, 95% humidity). Afterward, the explants were subjected to a single impact load of 0.59 J using a drop-tower model as previously described (*Riegger and Brenner, 2019*; *Riegger et al., 2016*; *Riegger et al., 2018a*) and further cultivated under serum-free conditions (DMEM, 1% sodium pyruvate, 0.5% L-glutamine, 1% non-essential amino acids, 0.5% penicillin/streptomycin, 10 µg/mL 2-phospho-L-ascorbic acid trisodium salt, and 0.1% insulin-transferrin-sodium selenite).

### Stimulation of isolated hAC

SIPS was induced by stimulation with 0.1 µM Doxo (Selleckchem, S1208) for 10 days (*Kirsch et al., 2022*). Activated Sim was applied for 20 hr at a concentration of 10 µM. Induction of apoptosis was attained by stimulation with cycloheximide (10 µg/mL; Selleckchem, S7418) and TNF (10 ng/mL; Peprotech, 300-01A) as previously reported (*Riegger and Brenner, 2019*). All stimulations were performed in serum-reduced chondrocyte medium (5% FBS).

### alamarBlue assay (cytotoxicity/cell proliferation assay)

Quantitative measurement of cell proliferation/cytotoxicity was attained by means of an alamarBlue assay (Bio-Rad, Munich, Germany). Cells were incubated for 3 hr in a 5% alamarBlue solution (in DMEM) at 37°C. Fluorescence intensities were detected at 550 nm excitation and 590 nm emission by using the multimode microplate reader Infinite M200 Pro (Tecan Austria GmbH Groedig, Austria). Blank values (5% alamarBlue solution in empty well) were subtracted from measured values. Results were calculated relative to unstimulated cells.

### Flow cytometric analysis

To analyze surface markers on freshly isolated hAC, cartilage tissue was directly digested by means of collagenase overnight, without pronase. In case of cultured hAC, cells were detached with PBS-buffered EDTA (5 mM). hAC were stained with anti-CSV (clone 84-1, abnova, #H00007431-MF08) or IgG2bκ isotype control (BD Pharmingen) for 40 min. A minimum of $2×10^4$ cells was analyzed on a Becton Dickinson FACSCalibur flow cytometer (BD Biosciences) with dual-laser technology using the corresponding software CellQuest version 5.2.1. A maximum of 1% isotype control-positive cells was accepted. Cancer cell lines HeLa (low CSV levels; mean: 11.8%±4.2) and SaOs-II (mediate/high

CSV levels; mean: 34.4%±8.8) were used to validate the anti-CSV antibody and the staining protocol (*Supplementary file 1*). While CVS was found to be degraded by trypsin and accutase treatment, no influence was observed in case of collagenase relative to EDTA (*Supplementary file 1*).

## SA-β-gal staining and activity assay

SA-β-gal staining was performed using an SA-β-gal staining kit according to the manufacturer's protocol (Cell Signaling Technology, Danvers, MA, USA). In short, cells were seeded on chamber slides (5000 cells/cm$^2$). Cells were fixed in a 2% formaldehyde and 0.2% glutaraldehyde solution for 15 min. After washing with PBS, cells were stained overnight in an X-gal staining solution at 37°C (dry incubator; low $CO_2$).

In case of the SA-β-gal activity assay, a fluorometric cellular senescence assay kit was used according to the manufacturer's protocol (Cell Biolabs Inc, San Diego, CA, USA). In short, cells were seeded on a 96-well plate (5000 cells per well). The next day, cells were lysed using the cell lysis buffer and centrifuged. The supernatants of the cell lysates were mixed with the assay buffer and incubated for 2 hr at 37°C in the dark. After stopping the reaction using the stop solution, fluorescence was measured on a black bottom 96-well plate at 360 nm excitation and 465 nm emission using an infinite M200 PRO TECAN reader.

## siRNA-mediated knockdown of CDKN2A or VIM

Adherent hAC were incubated in a digest solution containing 1 mg/mL collagenase, 1 mg/mL protease, and 40 U/mL hyaluronidase to increase siRNA transfection efficacy. Detached cells were counted and seeded at a density of 20,000 cells/cm$^2$. Silencer Select siRNA (Thermo Fisher Scientific) against CDKN2A (ID: s218; targeting both p16$^{INK4a}$ and the related transcript p14$^{ARF}$), VIM (ID: s14800), or Silencer Negative Control siRNA was diluted in Lipofectamine P3000 (Thermo Fisher Scientific) and added to the cells at a final concentration of 5 pmol/μL in serum-free Opti-MEM (Thermo Fisher Scientific; 31985062) without antibiotics. After 4 hr at 37°C, transfection medium was removed and serum-containing chondrocyte medium was added. After 72 hr, hAC were trypsinized, counted, and seeded as mentioned above. The next day, transfection was repeated in adherence. This protocol allows a consistent silencing over 7 days.

## DCFDA and MitoSOX assay

Detection of cytoplasmatic and mitochondrial ROS levels was performed by means of a commercial DCFDA/H2DCFDA – Cellular ROS Assay Kit (abcam, ab113851) and MitoSOX Red Mitochondrial Superoxide Indicator (Invitrogen, Thermo Fisher Scientific, M36008), according to the manufacturer's instructions.

## Quantitative real-time PCR

For total RNA isolation, cryopreserved cartilage explants were pulverized with a microdismembrator S (B. Braun Biotech, Melsungen, Germany). Subsequently, RNA was isolated using a Lipid Tissue Mini Kit (QIAGEN, Hilden, Germany). In case of isolated cells, mRNA of least 50,000 cells was isolated using a RNeasy Mini Kit (QIAGEN). RNA was reverse transcribed with Superscript II (Thermo Fisher) and used for qRT-PCR analysis (StepOne-PlusTM Real-Time PCR System, Applied Biosystems, Darmstadt, Germany). Relative expression levels were calculated using the ΔΔCt method, except of correlation analysis depicted in *Figures 2j–4n*, in which the ΔCt method was required. GAPDH, HPRT1, and TMEM199 served as housekeeping genes. TMEM199 has previously been described as appropriate endogenous control to study gene expression in senescent chondrocytes (*Kuwahara et al., 2020*), while GAPDH or HPRT1 are the most commonly used housekeeping genes in studies on chondrosenescence.

## Immunocytochemistry/fluorescence staining

Isolated cells were fixed with formalin, blocked for 1 hr at 37°C, and stained with anti-CSV (clone 84-1, abnova, H00007431-MF08 or H00007431-MB08), anti-VIM (clone EPR3776, abcam, ab185030), and anti-CDKN2A (abcam, ab108349) for 2 hr at RT. In case of intracellular staining, cells were permeabilized with 0.1% PBS-Tween 20. Unconjugated anti-CDKN2A was further incubated with a biotinylated link antibody (DAKO), followed by iFluor568-conjugated avidin (ATT Bioquest), each for 20 min at

RT. In case of immunocytochemistry, anti-CSV H00007431-MB08 was incubated as described above, followed by further incubated with strepavidin-bound HRP (DAKO) for 20 min each. After addition of AEC substrate for 15 min, nuclei were counterstained with hematoxylin.

## In vitro differentiation

For chondrogenic differentiation, chondrocytes were cultivated for 28 days in CDM (DMEM 4.5 g/L glucose and Ham's F12 (1:1), 100 U/mL penicillin/streptomycin, 40 ng/mL L-proline, 2 mM L-glutamine, 1 mM Na-pyruvate, 0.1 µM dexamethasone, 50 µg/mL ascorbic acid, 10 µL/mL ITS [Sigma-Aldrich], 10 ng/mL rhTGF-β3 [Peprotech], 10 ng/mL rhBMP6 [Peprotech]) under serum-reduced (in case of 2D culture; 1% FSC) or serum-free (in case of pellet culture) conditions as previously described (*Riegger et al., 2018b*). For osteogenic differentiation, chondrocytes were cultured for 28 days in ODM (DMEM 1 g/L glucose, 10% FBS, 100 U/mL penicillin/streptomycin, 2 mM L-glutamine, 0.1 µM dexamethasone, 0.2 mM ascorbic acid, and 10 mM β-glycerophosphate).

## Statistical analysis

Experiments were analyzed using GraphPad Prism8 (GraphPad Software, Inc, La Jolla, CA, USA). Data sets with n≥5 were tested for outliers by means of the Grubbs outlier test. Outliers were not included in statistical analyses. Each data point represents an independent biological replicate (donor); technical replicates were not performed. Significance level was set to α=0.05. Information about the statistical analysis applied is provided in the corresponding figure legends.

## Acknowledgements

The authors would like to thank Natalie Braun for excellent technical assistance and Hartmut Geiger for his constructive input. This research study was supported by the European Social Fund and by the Ministry of Science, Research and Arts Baden-Württemberg, as well as the University of Ulm (Bausteinprogramm). Jana Riegger Margarete von Wrangell fellowship (European Social Fund and by the Ministry of Science, Research and Arts Baden-Württemberg); start-up funding of the University of Ulm (Bausteinprogramm). The funders had no role in study design, data collection and interpretation, or the decision to submit the work for publication.

## Additional information

### Funding

| Funder | Grant reference number | Author |
|---|---|---|
| European Social Fund | | Jana Riegger |
| Ministry of Science, Research, and Arts Baden-Württemberg | | Jana Riegger |
| University of Ulm | | Jana Riegger |

The funders had no role in study design, data collection and interpretation, or the decision to submit the work for publication.

### Author contributions

Jana Riegger, Conceptualization, Data curation, Formal analysis, Funding acquisition, Investigation, Visualization, Methodology, Writing – original draft; Rolf E Brenner, Resources, Data curation, Writing - review and editing

### Author ORCIDs

Jana Riegger https://orcid.org/0000-0003-0048-5047

### Ethics

Human cartilage was obtained from donors undergoing total knee joint replacement due to OA. Informed consent was obtained from all patients according to the terms of the Ethics Committee of the University of Ulm (ethical approval number 353/18).

### Decision letter and Author response

Decision letter https://doi.org/10.7554/eLife.91453.sa1
Author response https://doi.org/10.7554/eLife.91453.sa2

---

## Additional files

### Supplementary files

MDAR checklist

Supplementary file 1. Figures S1-S8.

### Data availability

All data generated or analysed during this study are included in the manuscript and supporting file.

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
