## [Editor Report]

The manuscript provides evidence for an association between cell surface vimentin (CSV) and chondrocyte senescence. The authors have provided solid in vitro evidence about the role of cell surface vimentin (CSV) in chondrocyte senescence and its potential in OA development, although it is lacking in vivo evidence to prove the conclusion. Overall, this is well suited for a broad audience and is important for the OA/chondrocyte biology community.

---

## [Decision Letter]

[Editors' note: this paper was reviewed by Review Commons.]

---

## [Author Response]

Point-by-point description of the revisionsReviewer #1 (Evidence, reproducibility and clarity (Required)):The manuscript by Rigger and Brenner details the role of vimentin network, in advancing OA pathogenesis by exacerbating premature senescence. The data is well presented and the study of interest, in that there is little known about vimentin in cartilage biology.The authors used OA derived cartilage explants and chondrocytes cultures, were graded for severity and compared accordingly. Figure 1 shows that markers of senescence are increased with structural damage, which is well established and consistant with the literature. Using a DOX model the authors induce premature senescence and exhibit a disrupted vimentin network. However, upon KD of CDKN2A, a marker of senescence, but did not observe complete reversal of CSV presentation.Next the authors show in figure 4 and 5, that the reduction or dismemberment of vimentin structures are linked to senescence and may act as contributing factors.Figures 6 and 7 then go on to show that upon advanced passage chondrocytes lose their vimentin network, and tend to senesce and mineralize.Reviewer #1 (Significance (Required)): Strength:This is a very novel study showing a link between vimentin and senescence in chondrocytes. The data are in line with other data. The work is clearly written structured and well displayed.

We thank reviewer #1 for their interest in our work and their overall positive report.

Suggestions for improvement:While the study is very thorough ought in describing the markers of senescence and vimentin network, it lacks insight regarding mechanism which isn't completely deciphered. Are there links to key transcription factors?

The transcriptional regulation of vimentin in human cells is very complex. The VIM promoter region comprises multiple elements, such as a NF-κB- binding site, a PEA3-binding site and two AP1-binding sites (Zhang et al., 2003). Moreover, it was recently demonstrated that redox signaling is involved in vimentin expression at the wound margin after tissue injury in zebra fish (LeBert et al., 2018). However, it has also been reported that IL-1ß stimulation results in reduced gene expression of vimentin via p38-signalling in cartilage degeneration and OA progression (see manuscript REF. 36,37).

In our study, we observed that enhanced CSV levels are associated with a decreased vimentin gene expression, indicating a lower stability of the mRNA or decreased transcription of VIM in senescent chondrocytes (maybe due to enhanced p38-signalling as mentioned above). Since the transcriptome in senescent cells is radically changed, this question cannot be answered easily.

In future studies, we will rather try to clarify the underlying mechanism of vimentin externalization. There are still many questions to be answered: is the CSV anchored in the cell membrane (which anchor protein?) and is there still a connection to the intracellular vimentin network? Which proteins are involved in the externalization process: maybe comparable to phosphatidylserine exposure, mediated by flippases, scramblases, and lipid transfer proteins or rather by vesicles?

Literature mentioned above (not included in manuscript):

LeBert et al., 2018: Damage-induced reactive oxygen species regulate vimentin and dynamic collagen-based projections to mediate wound repair. DOI: 10.7554/eLife.30703

Zhang et al., 2003: ZBP-89 represses vimentin gene transcription by interacting with the transcriptional activator, Sp1. DOI: 10.1093/nar/gkg380

It is also unclear if disruption of the network is more detrimental than KD in promoting senescence.

KD of Vimentin led to a gradually decrease of intracellular Vimentin content and consequent stress. The cells were analyzed 7 days after induction of the KD and exhibited a stable senescent phenotype, comparable to Doxorubicin-treated chondrocytes (treated with very low concentrations over several days to produce only mild but ongoing stress). These models might reflect the pathophysiologic situation: We think that cellular stress due to mechanical impact and subsequent oxidative stress/ low-grade inflammation might lead to a gradual disruption or re-organization of the vimentin network, which is accompanied by decreased vimentin gene expression.

In case of the disruption of the vimentin network by Simvastatin, the stress response was very intense and rapid (24 h), and was only conducted as a proof-of-principle experiment. Despite the upregulation of some senescence-associated markers, we don`t think that permanent Simvastatin treatment would be suitable to obtain a stable senescent phenotype, but rather expect the cells to die due to excessive stress.

It would have been good to include models OA murine models to understand these processes better, and make a stronger physiological connection with OA of the joint.

The CSV antibody is only suitable for human cells and cannot be used for immunohistochemistry. Therefore, all previous reports of CSV are based on human (isolated) cells. At the current time point, it would not be possible to stain CSV in joints of mice after induction of PTOA due to the methodological limitations. We actually tested the CSV-antibody in isolated lapine chondrocytes and found a high percentage of CSV-positive cells, even at low passages. Although stress increased the amount of CSV-positive lapine cells, we did not consider the results as reliable due to the high percentage in un-stressed cells, which might result from unspecific antibody binding.

Overall, we think that the usage of clinical OA samples is convincing and reflect the pathophysiologic situation in the human OA joint.

Reviewer #2 (Evidence, reproducibility and clarity (Required)):The manuscript provides solid evidence for an association between cell surface vimentin (CSV) and chondrocyte senescence. Human cartilage and cultured chondrocytes are used with a wide range of approaches to provoke senescence: natural osteoarthritis, traumatic loading ex vivo, doxorubicin to cells in monolayer, vimentin siRNA, and simvastatin. In contrast, relatively little was done to try and interrupt or reverse the role of CSV in senescence, with CDKN2A siRNA representing one attempted intervention. The manuscript is well written and the data are presented in a logical and clear manner, with a high likelihood of being reproduced in subsequent studies.

We thank reviewer #2 for their interest in our work and their mainly positive report.

Regarding their comment on our attempts to reverse CSV on senescent chondrocytes, we would like to add the following: Reversal of cellular senescence is a very ambitious challenge. But in fact, we are currently preparing a manuscript in which we characterize an appropriate senolytic strategy to “rejuvenate” human chondrocytes and plan to use this approach to reduce the amount of senescent and thus CSV-positive cells in future experiments.

Major comments:In the doxorubicin experiments, the senescent cells show a spread morphology as expected. Given the importance of vimentin in cell spreading (as the authors own data show), the possibility that spread morphology itself (and not senescence) leads to CSV should probably be examined. This could perhaps be achieved by plating with different concentrations of fibronectin or other matrix proteins that produce a spread morphology to a degree that matches the doxo. If the cells remain spread for ~10 days but don't become senescent and don't have CSV, this would provide further support for a direct relationship.

We agree that cell spreading is associated with various cellular processes (for example by the YAP signaling pathway). Moreover, we would like to thank the reviewer for the proposed experiment.

Seeding of cartilage cells on fibronectin coated plates is a commonly used procedure to isolate chondrogenic stem progenitor cells, due to their higher affinity to fibronectin. The cells are usually cultured for several days on the coated plates and do not exhibit a flattened, senescent-like phenotype (as we observe for Doxorubicin-treated cells), but an elongated, fibroblast-/ stem cell-like shape. Our results (Figure 6E) demonstrate that CSPC have no increased CSV levels, despite their elongated (not flat) morphology.

There are some findings supporting the assumption that CSV leads to enhanced cell adhesion, but not that adhesion or cell spreading promotes CSV: we included experiments with HeLa (low CSV levels) and SaOS-2 (high CSV levels), which demonstrated that high CSV levels are associated with increased plastic adhesion (Figure S5). In line with this, we demonstrated that higher CSV levels on chondrocytes were associated with enhanced fibronectin and vitronectin binding, which might explain increased plastic adhesion. Moreover, Simvastatin stimulation and subsequent cellular stress by Vimentin disruption resulted in enhanced CSV but did not lead to cell spreading (Actin not affected, cells rather elongated, not flattened).

Minor comments:The CSV antibody and staining method appeared to have generated some signal from debris, which makes it challenging to assess the localization of true staining. Presumably the true staining would be present only on the cell surface. While the widefield view is appreciated, perhaps insets with a higher magnification would clarify.

In Figure 2h and Figure 2i, we provide insets of the IF-staining and an exemplary image made by scanning electron microscopy (SEM). CSV is not localized on debris – Figure 2h, actually represents the cell surface. The magnified, Doxo-treated cell is highly senescent and thus flattened. The uneven (rather spotted) staining pattern of CSV and the unusual shape of the cell might suggest that this is debris, not the cell membrane.

For figure 1k, it is a bit surprising that CDKN2A would peak so early after injury and then drop off. Most studies in other systems show a gradual increase in CDKN2A levels with persistent stress as opposed to a rapid increase in response to acute stress. Could the drop-off be due to preferential death of these cells? The CSV % in 1m was taken from 7d after trauma (plus 7 days in monolayer it appears). Further discussion on the timing of traditional senescence markers as compared to the emergence of CSV would be useful.

We would like to thank the reviewer for this comment. That CDKN1A was induced by mechanical trauma without significant decrease at the later time points was in line with the P53 expression, which we detected via immunohistochemistry (IHC; positive staining of chondrocyte nuclei in cartilage). P53 and P21 are regarded as interconnected senescence markers. Interestingly, P53 is not regulated on gene expression level upon cartilage trauma or Doxorubicine stimulation – but there is a significant increase in P53 nuclear translocation.

Although such a discrepancy between gene expression and protein activity has not been reported in case of P16 or P21, we plan to investigate the dynamics of these cell cycle regulators and its connection to CSV after cartilage trauma in more detail in future studies.

We included the following statement in the discussion part:

“In the current study, we observed that CSV on chondrocytes was reduced by siRNA-mediated silencing of CDKN2A and increased after Doxo treatment or cartilage trauma. While we confirmed that mRNA levels of both CDKN1A and CDKN2A were significantly enhanced upon injury but exhibited different expression levels over time, we determined CSV-positive cells only at one time point after ex vivo cartilage trauma. Future studies might also consider earlier and later time points after cartilage injury to identify a potential time-dependent peak or decline in CSV-positive chondrocytes. In this way a potential association between CSV and the expression levels of CDKN1A and CDKN2A, which are thought to play differential roles in initiating and maintenance of senescence, respectively [50], might be clarified.”

[50] Stein G, Drullinger L, Soulard A, and Dulić V. Differential Roles for Cyclin-Dependent Kinase Inhibitors p21 and p16 in the Mechanisms of Senescence and Differentiation in Human Fibroblasts. Mol Cell Biol. 1999;19(3): 2109–2117. https://doi.org/10.1128/mcb.19.3.2109.

There is no CSV staining shown for figures 4 and 5. While the quantification of CSV was done by flow cytometry, it would nice confirmation to see the increase in CSV on the surface of cells with either siRNA for vimentin or the simvastatin.

There is no CSV staining shown for figures 4 and 5. While the quantification of CSV was done by flow cytometry, it would nice confirmation to see the increase in CSV on the surface of cells with either siRNA for vimentin or the simvastatin. CSV-IF of simvastatin-treated chondrocytes is provided in Figure 5 (b). We did not perform exemplary staining of CSV after VIM-KD, because the quantification was performed via flow cytometry.

Reviewer #2 (Significance (Required)):The strengths of the study include a rigorous design and the establishment of a potential new cell surface marker of chondrocyte senescence. The main limitation is that the conclusions are largely descriptive in nature.If CSV is confirmed as a robust marker of senescence, this would be of value to the field. While this marker has been explored previously in other systems, there is value in this manuscript given the wide range of contexts investigated for a cell type in which senescence likely has an important role.Reviewer #3 (Evidence, reproducibility and clarity (Required)):This study presents a sound piece of science in the puzzle about extracellular vimentin in the differentiation/dedifferentiation of human chondrocytes and senescence and osteoarthritis. Eventhough, no mechanism is elucidated, the results clearly point towards a correlation of the amount of extra cellular vimentin and the level of chondrocyte senescence, and therefore signs of osteoarthritic changes in the cultivated chondrocytes. The methods applied are state-of-the art and provide the means to generate meaningful results in this experimental setting. The paper is concise and clearly written, there are only minor remarks.

We thank reviewer #3 for their interest in our work and their overall positive report.

Minor comments: 1. The main clue of the paper is extra cellular vinemtin around chondrites in culture, please provide better pictures (1g) to support this. Why is the extra cellular staining seen so broad and not concentrated on the cells surface? The picture chosen imply a huge amount of vimentin to be externilized in disease states. It also indicates that in diseased chondrocytes no intact or semi-intact vimentin network is found intracellular. Please comment.

In Figure 1g, CSV is located on the cell membrane. The pattern of the staining was surprising to us, as well. CSV was not equally distributed on the membrane, but rather represented an inconsistent pattern. Sometimes the staining was located at the filopodia of the cells, sometimes the whole cell was covered by spots. We also observed this on cancer cells, which was in line with other studies using this antibody. It remains unclear whether the distribution of the CSV has any effect. But we assume that the high abundance in filopodia might be connected with cell adhesion and mobility, which was positively associated with CSV.

Yes, chondrocytes isolated from highly degenerated tissue exhibited higher CSV levels as compared to cells derived from macroscopically intact regions. Although we did not investigate the vimentin network of these cells, our observations in Doxo-treated cells imply, indeed, that intracellular vimentin might be altered in diseased chondrocytes. According to this, Blain et al. (Ref. 13) reported that there is a disassembly of the intracellular vimentin network in OA chondrocytes, which can disturb the chondrocyte phenotype and contributes to the development of OA (see discussion).

2. In the doxo experiment no extracellular vimentin is found? Please explain.

Doxo-treated cells are highly positive for CSV (=extracellular vimentin on membrane). However, the intracellular vimentin is strongly decreased and some cells seem to be negative. We have not clarified the underlying mechanism by now, but it seems that senescence/ disease progression negatively affects the transcription of vimentin and, at the same time, promotes the externalization of the existing intracellular vimentin. Altogether, this might result in a decline in intracellular vimentin.

3. The SEM picture is showing what. IGH? The red dots are colloidal gold particles? In any case the quantity of stain gathered EM level would not correlate to the huge amount seen in LM staining. Please comment.

For the SEM analysis, a gold particle-coated secondary antibody was used. The positive signal usually appears in white and was subsequently colored via a software. In IF and ICC staining, we had a signal amplification due to the biotin-streptavidin system and the magnification makes, of course, a huge difference.

4. Why the ICC in Figure 3c? The siRNA is not detected in the KD? A reduction of Vimentin could be shown via WB.

In Figure 3c, the KD of P16 was confirmed on protein level. In addition to the gene expression analysis, we chose the ICC (IF) to confirm that there is a decline in active (nuclear) CDKN2A. In case of P53, we made the experience that gene expression and the amount of cytoplasmic/ nuclear protein might not be consistent.

In Figure 4, we confirmed the successful KD of vimentin on mRNA and protein level (flow cytometry plus IF). Of course, WB would also be possible, but we decided to use the methods in which the antibody was well established and we wanted to visualize the disturbance of the intracellular vimentin network upon KD.

5. Figure 4c, why are there no remnants of the vimentin networks seen in the chondrocytes? A Knock-down, not a KO is shown.

In fact, most of the intracellular vimentin seems to be gone. However, there are some remnants (condensed fibers/ bundles) of the former vimentin network. We applied the VIM-KD over seven days. Usually, a KD experiment is only conducted for 2-3 days. But since we were not sure how stable the vimentin protein would be, we chose seven days. This long-lasting KD might have resulted in a strong decline of the protein. Moreover, the CSV levels on these cells were very high, indicating that existing vimentin was externalized and additionally decreased the amount of intracellular vimentin.

6. Please comment of the concentration of simvastatin, why not nmolar?

The concentration of Simvastatin was chosen in accordance with Trogden et al. (Ref. 26), who first described the effects of simvastatin on the vimentin network. A lower concentration might have had the advantage, that the effects were less severe, allowing a longer observation time than 24h. However, as a proof-of-principle model to demonstrate the connection between vimentin network collapse ant CSV expression, the concentration worked quite well.

7. CSV+ is misleading in Figure 6g, it's not an over expression.

We would like to thank the reviewer for this comment and removed the “+” to make it less misleading.

8. The concept of EMT is debatable, at least in kidney fibrosis, and chondrocytes are not epithelial cells. Please add a more critical discussion point.

The authors agree with the reviewer’s argument that chondrocytes are no epithelial cells ant that the term EMT doesn’t seem to be appropriate. However, this is one leading hypothesis proposed by the working group of Prof. Mayán, who described CX43 and other EMT-markers on/ in senescent chondrocytes (see reference 31; more recently: Cell Death Dis. 2022;13(8):681. doi: 10.1038/s41419-022-05089-w).

We added the following passage in the discussion part to indicate that this hypothesis is a controversial concept:

“Nevertheless, the hypothesis that chondrocytes might undergo an EMT-like process remains controversially discussed, because chondrocytes are mesenchymal and not epithelial cells. In a recent review, Gems and Kern propose to consider senescent chondrocytes as activated and hyperfunctional remodeling cells occurring during OA progression [49]. Accordingly, chondrosenescence might represent an unsuccessful attempt of tissue repair. They further suppose that the senescent or activated chondrocytes are associated with a hypertrophic, bone-forming phenotype, following the process of bone development rather than hyaline cartilage formation. In line with this, we observed that CSV was associated with enhanced osteogenic capacities and a decline in chondrogenic properties.”

[49] Gems and Kern, 2022: Geroscience. 2022;44(5):2461-2469. doi: 10.1007/s11357-022-00652-x.

Reviewer #3 (Significance (Required)):The manuscript provides novel insight in the role of intermediary filaments, i.e. vimentin, on chondrocyte senescence and osteoarthritic changes in vitro. It's strength is a thorough elucidation of the connection with a wealth of experimental data, a weakness is the missing elucidation, or first experiments in the direction, of the cell biological mechanism.It is well suited for a broad audience, because it deals with fundamental cell biological phenomena, definitely it's important for the OA /chondrocyte biology community.